# Think before Recommendation:
# Autonomous Reasoning-enhanced Recommender

**Xiaoyu Kong**[1] **Junguang Jiang**[1] **Bin Liu**[1] **Ziru Xu**[1]
**Han Zhu**[1] **Jian Xu**[1] **Bo Zheng**[1] **Jiancan Wu**[3,4]* **Xiang Wang**[2]*
[1]Taobao & Tmall Group of Alibaba, China
[2]National University of Singapore
[3]Institute of Dataspace, Hefei Comprehensive National Science Center
[4]Shanghai Key Laboratory of Data Science
linlin.kxy@alibaba-inc.com
wujcan@gmail.com
xiangwang1223@gmail.com

## Abstract

The core task of recommender systems is to learn user preferences from historical user-item interactions. With the rapid development of large language models (LLMs), recent research has explored leveraging the reasoning capabilities of LLMs to enhance rating prediction tasks. However, existing distillation-based methods suffer from limitations such as the teacher model's insufficient recommendation capability, costly and static supervision, and superficial transfer of reasoning ability. To address these issues, this paper proposes RecZero, a reinforcement learning (RL)-based recommendation paradigm that abandons the traditional multi-model and multi-stage distillation approach. Instead, RecZero trains a single LLM through pure RL to autonomously develop reasoning capabilities for rating prediction. RecZero consists of two key components: (1) "Think-before-Recommendation" prompt construction, which employs a structured reasoning template to guide the model in step-wise analysis of user interests, item features, and user-item compatibility; and (2) rule-based reward modeling, which adopts group relative policy optimization (GRPO) to compute rewards for reasoning trajectories and optimize the LLM. Additionally, the paper explores a hybrid paradigm, RecOne, which combines supervised fine-tuning with RL, initializing the model with cold-start reasoning samples and further optimizing it with RL. Experimental results demonstrate that RecZero and RecOne significantly outperform existing baseline methods on multiple benchmark datasets, validating the superiority of the RL paradigm in achieving autonomous reasoning-enhanced recommender systems. Our codes are available at https://github.com/AkaliKong/RecZero.

## 1 Introduction

Recommender models aim to learn user preferences on items from historical user-item interactions (*e.g.,* ratings, clicks, purchases) [1–7]. Motivated by the rapid progress of large language models (LLMs) [8–11], recent research has explored adapting LLMs as recommenders, leveraging their strengths in world knowledge, semantic understanding, and reasoning. A promising direction is to leverage LLMs' reasoning capabilities [12–16] to enhance rating prediction [17–19] — a core recommendation task that explicitly models user preferences by predicting their ratings on items.

Upon scrutinizing prior studies on such reasoning-enhanced rating predictors [20–25], we can summarize a common pipeline: (1) Given a user's rating on a target item, a teacher model — often a

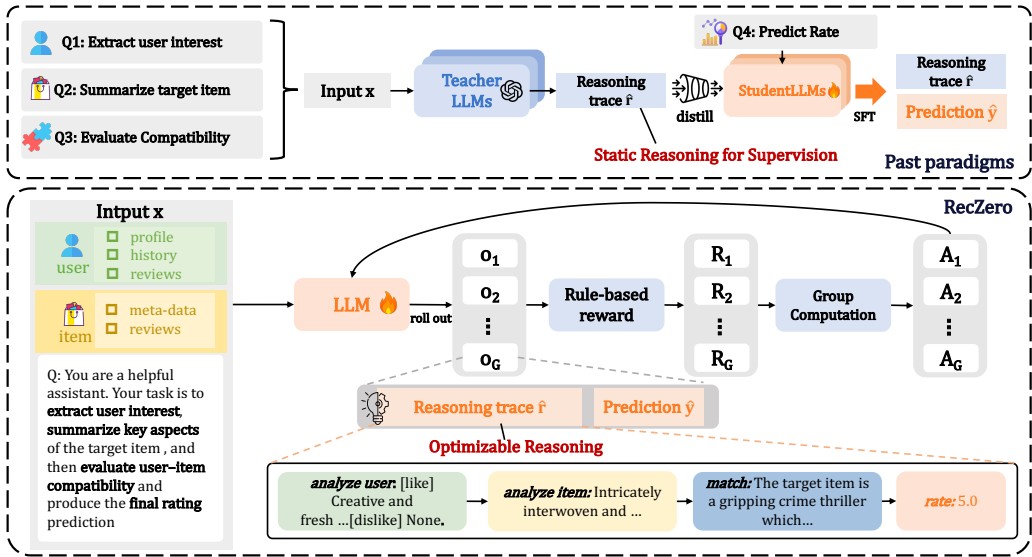

Figure 1: Comparison between RecZero and Conventional Paradigms. Conventional paradigms train multiple student LLMs using diverse query-based datasets with reasoning traces, while RecZero uses pure RL to train a single LLM for the entire workflow. Unlike conventional methods reliant on Teacher models, RecZero leverages Reward Signals to jointly optimize reasoning traces and predictions, improving efficiency and performance.

strong general-purpose LLM such as ChatGPT [8] — first extracts the user's preferences over item attributes from historical interactions, then generates possible intermediate reasoning steps to interpret the observed rating; (2) Each reasoning trace, paired with the corresponding user rating, forms an instruction sample; (3) A smaller model is then supervised fine-tuned on these reasoning-enhanced instruction samples to distill the teacher model's extraction and reasoning capabilities.

Despite recent progress, we identify several limitations inherent in this distillation paradigm, as evidenced in Figure 1:

- **Limited Recommendation Capability of the Teacher.** General-purpose teacher models lack domain-specific knowledge for recommendation, producing reasoning traces that are misaligned with the rate prediction objective. Consequently, these suboptimal reasoning processes hinder the student's performance.

- **Costly and Static Supervision.** Generating high-quality reasoning data at scale, whether via human annotation [26] or LLM API calls [27, 28], is both time- and resource-intensive. Moreover, the student model is restricted to passively consuming static teacher outputs, with no chance to actively refine or optimize the reasoning process.

- **Superficial Transfer of Reasoning Ability.** While the student model fine-tuned on teacher-generated data can reproduce reasoning traces, it often imitates surface-level patterns, rather than acquiring genuine reasoning skills [29]. This results in overfitting to the distribution of teacher instructions, limiting generalization to unseen tasks [29, 30].

To address the limitations of existing distillation-based methods, we draw inspiration from the recent success of reinforcement learning (RL) in LLM post-training [31–33], particularly DeepSeek-R1-Zero [32], and propose a simple yet effective RL paradigm for LLM-based recommendation, dubbed **RecZero**. Discarding the distillation paradigm that requires different models and disjoint stages (*i.e.*, information extraction and reasoning process generation of the teacher model, supervised fine-tuning of the student model), RecZero trains a single LLM via pure RL to develop autonomous reasoning capabilities for rating prediction. It is composed of two key components:

- **"Think-before-Recommendation" Prompt Construction.** Beyond the standard prompt containing a user's history and a target item, we introduce a structured reasoning template with special tokens to elicit step-wise analysis: "`<analyze user> ... </analyze user>`" prompts the model

to extract user interest from her/his historical interactions; "`<analyze item>` ... `</analyze item>`" summarizes key aspects of the target item; "`<match>` ... `</match>`" evaluates user–item compatibility based on the analyzed information; and consequently "`<rate>` ... `</rate>`" produces the final rating prediction. It decomposes the rating prediction task into discrete steps and employs chain-of-thought reasoning.

- **Rule-based Reward Modeling.** We adopt group relative policy optimization (GRPO) [31] to optimize the LLM with rule-based rewards. Given a prompt, the model samples multiple reasoning trajectories. For each trajectory, the reward is computed as the difference between the ground-truth rating and the predicted rating in the `rate` step. For the group of trajectories, we derive the relative advantage over the average reward as the signal to optimize the LLM.

As a result, unlike distillation-based methods that risk overfitting to surface-level reasoning traces, RecZero enables the LLM to acquire reasoning ability through interaction and optimization. The unified reasoning steps — `<analyze user>`, `<analyze item>`, `<match>`, and `<rate>` — are jointly trained to reflect and refine toward better recommendation performance, guided by recommendation-specific objectives. Beyond the pure RL approach of RecZero, we also explore a hybrid paradigm inspired by RL with cold start [32], termed **RecOne**. RecOne first constructs a small set of cold-start reasoning samples tailored for recommendation tasks, which are used to initialize the LLM via supervised fine-tuning. This warm-start model is then further optimized using RL to enhance its reasoning capabilities. By combining data-efficient initialization with task-specific RL, RecOne aims to achieve faster convergence and stronger performance in recommendation reasoning. We assess the effectiveness of RecZero and RecOne through extensive experiments on four benchmark datasets (*e.g.,* Amazon-book, Amazon-music [34], Yelp [35], IMDb [36]), showcasing the RL paradigm's superiority over distillation methods (*e.g.,* Rec-SAVER [20], EXP3RT [22], Reason4Rec [21]).

## 2 Preliminary

### 2.1 Reasoning-enhanced Rating Prediction

The primary goal of rating prediction task is to predict the user's ratings for items that align with user preferences. Let $\mathcal{U}$ denote a set of users, $\mathcal{I}$ a set of items, and $\mathcal{Y}$ the set of possible ratings. Formally, consider a user $u \in \mathcal{U}$ with a historical interaction sequence represented as $\mathcal{H}_u = <h_{u,1}, h_{u,2}, ..., h_{u,t}>$, where $h_{u,i} = (\mathcal{M}_i, y_{u,i}, d_{u,i})$ is a triplet that includes the meta-information $\mathcal{M}_i$ of the item $i$, the user's rating $y_{u,i} \in \mathcal{Y}$ of the item, and the user's review $d_{u,i}$. The prediction rule is defined as follows:

$$\hat{y}_{u,i} = \underset{k \in \mathcal{Y}}{\arg\max} \mathbb{P}_\theta(y_{u,i} = k | \mathcal{H}_u, \mathcal{M}_i) \tag{1}$$

where $\hat{y}_{u,i}$ denotes the predicted rating for user $u$ on item $i$, $\theta$ represents the model parameters.

In the context of reasoning-enhanced rating prediction, LLM parametered by $\theta$ receives the interaction history $\mathcal{H}_u$ of user $u$ and the meta-information $\mathcal{M}_i$ of the item $i$ as input $x_{u,i}$. Subsequently, the LLM performs an explicit reasoning process and makes the final user rating prediction, expressed as:

$$\hat{r}_{u,i}, \hat{y}_{u,i} = LLM(x_{u,i}) = LLM(\mathcal{H}_u, \mathcal{M}_i) \tag{2}$$

where $\hat{r}_{u,i}$ is introduced as the reasoning trace and $\hat{y}_{u,i}$ represents the predicted rating. In our framework, the reasoning process is decomposed into analyzing user preferences and extracting the characteristics of the target item, followed by a matching analysis.

## 3 Methodology

To address the major challenges inherent in existing reasoning-enhanced rating prediction methods, we propose RecZero. This innovative approach leverages pure reinforcement learning to encourage the model to jointly optimize the four critical processes of user analysis, item analysis, matching, and rating under a unified recommendation-specific objective. To further enhance the model's performance in recommendation tasks, we introduce RecOne, building upon RecZero. RecOne employs a specialized sampling method to obtain high-quality reasoning traces, which are used to perform SFT for cold-starting the initial model. Subsequently, we conduct task-specific RL to achieve faster convergence and more robust recommendation performance, as illustrated in Figure 2.

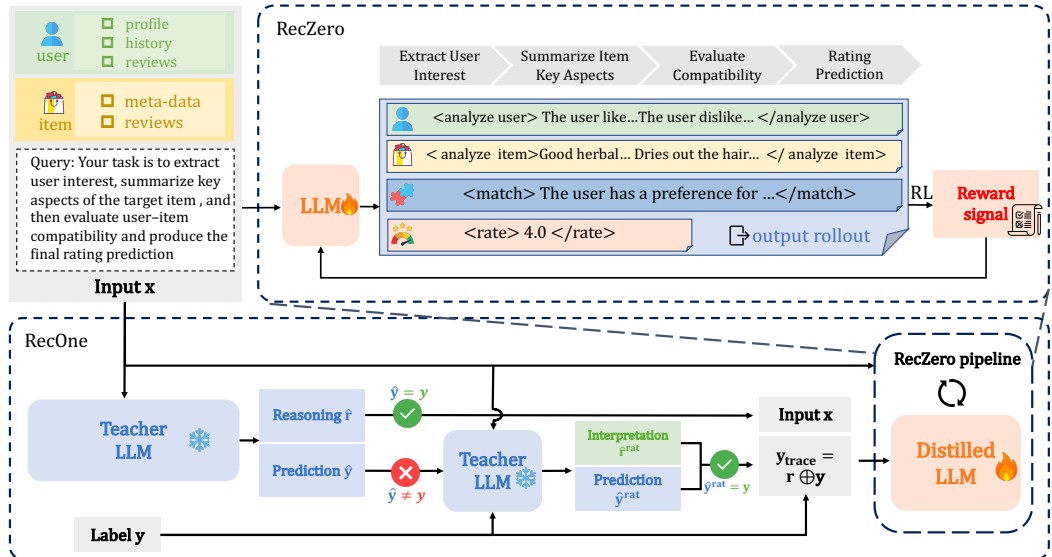

Figure 2: The RecZero and RecOne frameworks utilize a unified target to optimize multi-step reasoning within a coupled structure, achieving reasoning-enhanced predictions specifically designed for recommendation scenarios.

## 3.1 RecZero: Pure RL Paradigm for Reasoning-enhanced Recommendation

### 3.1.1 "Think-before-Recommendation" Prompt Construction

Previous studies [21, 22] have demonstrated that in reasoning-enhanced recommendation scenarios, requiring LLMs to first explicitly extract user preferences and item features, and then deliberate on the match between users and items, can significantly enhance the accuracy of LLMs in predicting users' actual ratings. High-quality rating predictions often rely on clear and high-quality extraction processes, whereas user preferences and item features typically lack explicit ground truth. To address this, we mandate that LLMs strictly output user preferences and item feature extractions according to predefined rules, and employ a rule-based reward mechanism to guide LLMs in self-optimization without the need for labeled data, thereby replacing the previous approach of using frozen LLMs with few-shot learning for these tasks. We have designed a system prompt to guide the LLM in generating reasoning trajectories beneficial to the rating prediction task. Specifically, we divide the reasoning process into three parts: user interest extraction, item aspects summarization, and compatibility evaluation. For a given user history and item metadata, we mandate that the LLM constrains the outputs of different steps within the specified token boundaries, as shown in Appendix.

### 3.1.2 Rule-based Reward Modeling

Rule-based approaches [31] can provide stable training rewards for RL. To address the issue of suboptimal performance caused by inconsistent training objectives in previously decoupled modules, we integrate user preference summarization, item feature distillation, and match deliberation within a unified framework and optimize them through a unified reward rule. For the format reward, we first define the correct format as follows:

1. The model's thinking process should be encapsulated within the `<analyze user>` ··· `</analyze user>`, `<analyze item>` ··· `</analyze item>`, and `<match>` ··· `</match>` tags, respectively.

2. The generated output must be within the `<rate>` ··· `</rate>` tags and free of any unreadable content.

Based on the above format requirements, the format reward is defined as follows:

$$R_{format} = \begin{cases} 0.5 & \text{if the format is correct} \\ -0.5 & \text{if the format is incorrect} \end{cases}, \tag{3}$$

In our training process, adhering to the continuous rating paradigm established in prior work [21, 22], we employ Rule-based Reward Modeling to encourage the model to approximate actual user ratings as closely as possible, rather than relying solely on correctness rewards. This objective can be formally expressed as:

$$R_{answer} = 1 - \frac{|y - \hat{y}|}{\text{max\_error}}, \tag{4}$$

where max_error is the maximum allowable error (which can be set based on the task, *e.g.,* when the rating range is from 1 to 5, max_error = 4). The final total reward can be formally expressed as $R = R_{format} + R_{answer}$. Unlike previous LLM-based paradigms, we do not design a separate task that requires the LLM to output an integer score that is then converted into a decimal rating via logit-weighted decoding. Compared with the SFT paradigm that imitates integer target labels, reinforcement learning offers a clear advantage: by crafting an appropriate reward function, we encourage the LLM to predict decimal ratings directly in its response, thereby earning a higher MAE reward and removing the need for the logit-weighted step altogether.

### 3.1.3 Group Relative Policy Optimization

We use Group Relative Policy Optimization (GRPO) [31] for RL policy optimization. In this approach, for a given input $x$[1], the method generates a set of outputs $\{y_1, y_2, \ldots, y_G\}$ using the existing policy $\pi_{\theta_{old}}$. The policy model is then refined by maximizing the following objective function:

$$
\begin{aligned}
J_{\text{GRPO}}(\theta) =& \mathbb{E}_{x \sim D, \{y_i\}_{i=1}^G \sim \pi_{\theta_{old}}(\cdot|x)} \left[ \frac{1}{G} \sum_{i=1}^G \frac{1}{|y_i|} \sum_{t=1}^{|y_i|} \left\{ \min \left( \frac{\pi_\theta(y_{i,t}|x, y_{i,<t})}{\pi_{\theta_{old}}(y_{i,t}|x, y_{i,<t})} \hat{A}_{i,t}, \right. \right. \right. \\
& \left. \left. \left. \text{clip} \left( \frac{\pi_\theta(y_{i,t}|x, y_{i,<t})}{\pi_{\theta_{old}}(y_{i,t}|x, y_{i,<t})}, 1 - \epsilon, 1 + \epsilon \right) \hat{A}_{i,t} \right) - \beta \mathbb{D}_{\text{KL}} \left[ \pi_\theta \| \pi_{\text{ref}} \right] \right\} \right],
\end{aligned}
\tag{5}
$$

where $\beta$ is a parameter that adjusts the trade-off between the task-specific loss and the KL-divergence. The advantage term $\hat{A}_i$ is derived from the rewards of the group of responses $\{R_1, R_2, \ldots, R_G\}$ and is calculated as:

$$\hat{A}_i = \frac{R_i - \text{mean}(\{R_1, R_2, \ldots, R_G\})}{\text{std}(\{R_1, R_2, \ldots, R_G\})}. \tag{6}$$

This formulation ensures that the policy optimization process is guided by the relative performance of the generated outputs within each group, promoting more stable and efficient learning.

## 3.2 RecOne: Distillation and RL Paradigm for Reasoning-enhanced Recommendation

While pure reinforcement learning approaches like RecZero show promise, we recognize opportunities for further enhancements. Specifically, the discrepancy between pretrained LLMs' training corpus and recommendation-specific data creates a domain gap that the RL paradigm must gradually bridge. Previous studies [37, 38] have demonstrated that LLMs can enhance their capabilities through self-generated reasoning processes. In this light, we propose a hybrid paradigm, RecOne, which first establishes a solid foundation through cold-start supervised fine-tuning on high-quality reasoning trajectories, then leverages the RecZero framework to further refine its reasoning capabilities.

### 3.2.1 Cold-Start Supervised Fine-Tuning

The first phase of RecOne addresses the foundational reasoning capabilities necessary for effective recommendations. We first perform a warm-up using recommendation data with long reasoning trajectories. Specifically, we sample $M$ data samples to construct a subset $D_{sub}$ from the complete dataset $D$, which is utilized in RecZero. For each data pair $(x, y) \in D_{sub}$, we leverage a teacher LLM (*e.g.,* DeepSeek-R1) with superior reasoning capabilities to generate a reasoning trajectory $(\hat{r}, \hat{y})$ given the input $x$, following Equation (2). When the predicted rating $\hat{y}$ aligns with the ground truth $y$, we consider the generated reasoning $\hat{r}$ to be high-quality; While for cases where $\hat{y}$ does not match $y$, we feed both $x$ and $y$ to the teacher model, requiring it to generate a rationalized trajectory $\hat{r}^{rat}$ that

---

[1]For notational clarity, we omit the user and item indices, while retaining other subscripts in the subsequent parts.

aligns with the correct rating. This process creates a reasoning trajectory dataset that consists of two complementary subsets:

$$D_{\text{trace}} = D_{\text{align}} \cup D_{\text{misalign}}, \tag{7}$$

$$D_{\text{align}} = \{(x, \hat{r} \oplus y) \mid \hat{y} = y\}, \tag{8}$$

$$D_{\text{misalign}} = \{(x, \hat{r}^{\text{rat}} \oplus y) \mid \hat{y} \neq y \wedge \hat{y}^{\text{rat}} = y\}. \tag{9}$$

Here, $D_{\text{align}}$ represents examples where the teacher LLM's reasoning naturally leads to the correct rating prediction, while $D_{\text{misalign}}$ captures cases requiring corrective reasoning to reach the proper conclusion. The training objective is formulated as an autoregressive model optimization problem:

$$\max_{\theta} \sum_{(x, y^{\text{trace}}) \in D_{\text{trace}}} \sum_{t=1}^{|y^{\text{trace}}|} \log P_{\theta}(y_t^{\text{trace}} | x, y_{<t}^{\text{trace}}), \tag{10}$$

where $y^{\text{trace}}$ represents the concatenation of reasoning trajectory and ground-truth rating as in Equations (8) and (9), $\theta$ denotes the LLM's model parameters, $y_t$ represents the $t$-th token in the output sequence, and $y_{<t}$ includes all preceding tokens in the sequence.

## 4 Experiments

In this section, we first demonstrate the performance improvement of the RecZero and RecOne framework on the rating prediction task. We requested the dataset from Reason4Rec [21] and conducted our experiments based on it. We conduct extensive experiments on real-world datasets, including Amazon-book, Amazon-music, and Yelp, to evaluate the effectiveness of the proposed RecZero and RecOne framework. Our analysis includes a detailed comparison of RecOne with existing baseline models, which cover CF-based models [39], Review-based models [40, 17, 18] and LLM-based recommendation models [20–22, 25].

We employed Mean Absolute Error (MAE) and Root Mean Square Error (RMSE) as evaluation metrics to reflect the model's accuracy in rating prediction. More details about these baseline models, datasets, and metrics can be found in Appendix. Additionally, we conducted comprehensive ablation studies to identify the key components that enhance the performance of RecOne, with a particular focus on the roles of the "thinking process" and the "format-guided model." Besides accuracy studies, we also compare the training and inference cost of RecZero against previous LLM-based paradigms. In short, we aim to address the following research questions:

- **RQ1:** How does RecZero and RecOne perform in comparison to other baseline methods?
- **RQ2:** Does the initial capability of the model have an impact on the upper limit of RL performance?
- **RQ3:** What is the impact of the designed components on the recommendation performance of RecOne?
- **RQ4:** How efficient is RecZero in terms of training and inference cost compared with the prior reasoning-enhanced baseline?

### 4.1 Performance Comparison (RQ1)

We conduct a holistic evaluation, considering metrics of both MAE and RMSE across Book, Music and Yelp datasets to demonstrate the effectiveness of our framework. This section comprehensively compares RecZero and RecOne against CF-based, Review-based and LLM-based baselines.

From Table 1 we observe that RecOne ranks first on all six evaluation metrics across the three domains. Concretely, it lowers the best previous RMSE by 6.7%, 12.2% and 6.2% on Book, Music and Yelp respectively, while cutting the MAE by 16.8%, 29.9% and 7.5%. When it relies solely on RL inside the LLM, with neither cold-start pre-initialization nor guidance from a teacher model, RecZero still surpasses all baselines in terms of MAE on the three datasets.

This section delves into the profound impact of the RecOne cold-start model on subsequent training processes. On the book dataset, we systematically compared the training trajectories of the distilled model obtained under the RecOne framework with those of the model trained using the initial model following the RecZero paradigm for RL, as illustrated in Figure 3.

Table 1: The Results of RecZero and RecOne compared with Traditional models and LLMs-based methods.

| Methods | Book | | Music | | Yelp | |
|---|---|---|---|---|---|---|
| | RMSE | MAE | RMSE | MAE | RMSE | MAE |
| **CF-based** | | | | | | |
| MF | 0.8565 | 0.6277 | 0.8142 | 0.6188 | 1.0711 | 0.7980 |
| **Review-based** | | | | | | |
| DeepCoNN | 0.8403 | 0.6211 | 0.8057 | 0.6034 | 1.0665 | 0.8312 |
| NARRE | 0.8435 | 0.6242 | 0.7881 | 0.5799 | 1.0785 | 0.8177 |
| DAML | 0.8371 | 0.6214 | 0.7848 | 0.5703 | 1.0405 | 0.7964 |
| **LLM-based** | | | | | | |
| Rec-SAVER | 0.9356 | 0.6645 | 0.9262 | 0.6463 | 1.1282 | 0.8295 |
| EXP3RT | 0.9346 | 0.6042 | 0.8312 | 0.5548 | 1.1420 | 0.8236 |
| Reason4Rec | 0.8325 | 0.5937 | 0.7647 | 0.5352 | 0.9972 | 0.7473 |
| **Ours** | | | | | | |
| RecZero | 0.8387 | 0.5253 | 0.7058 | 0.4271 | 1.0521 | 0.7429 |
| RecOne | **0.7784** | **0.5017** | **0.6776** | **0.3816** | **0.9774** | **0.7012** |

## 4.2 In-Depth Analysis of RecZero and RecOne (RQ2)

Notably, the initial model of RecZero exhibited an anomalous increase in MAE during the first 20 training steps, primarily attributed to the initial LLM focusing on acquiring relatively easier-to-obtain format rewards. In contrast, the distilled model of RecOne demonstrated higher format scores at the onset of training, enabling it to optimize the answer scores for the rating task more rapidly during RL training. Throughout the displayed 200 training steps, the RecOne model consistently outperformed RecZero, a result that unequivocally demonstrates the effectiveness of the cold-start strategy in enhancing the model's initial capabilities, thereby significantly elevating its performance ceiling for RL optimization. During the early stages of RL training, the MAE drops sharply, then decreases slowly with minor fluctuations, and finally stabilizes over a longer training period.

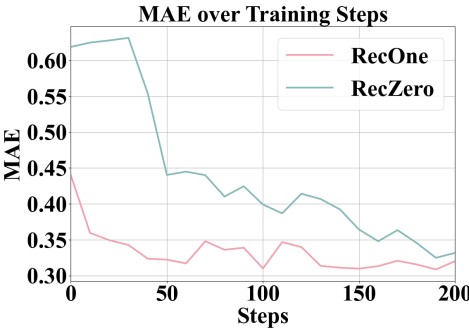

Figure 3: The performance of RecZero and RecOne over training steps.

## 4.3 Ablation Study (RQ3)

To validate the effectiveness of each component in the Rec-zero framework, we compare it with the following alternative approaches:

- **No Thinking Process:** We eliminate the required thinking process in the model output, requiring the model to directly predict the specific rating value based on the input.

- **No Multi-step Thinking:** We replace the multi-step thinking process `<analyze user>`, `<analyze item>` and `<match>` with a single-step `<think>` without imposing specific format requirements.

- **Correctness Reward Only:** We use a single correctness reward as the reward signal. Specifically, when the predicted score exactly matches the target, a correctness reward of 2 points is given; otherwise, it is assigned 0.

- **Only SFT for warm-start:** We discard the step in Rec-one that jointly optimizes reasoning and rating prediction through reinforcement learning, and instead only use the Teacher Model to generate trace data for cold-start scenarios.

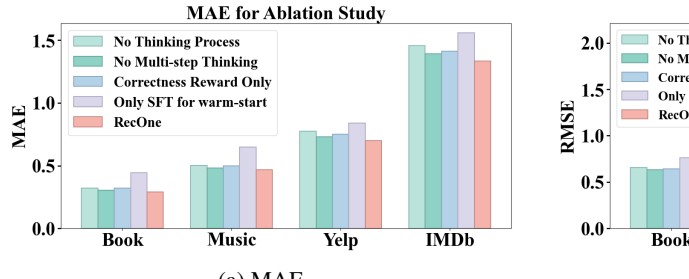
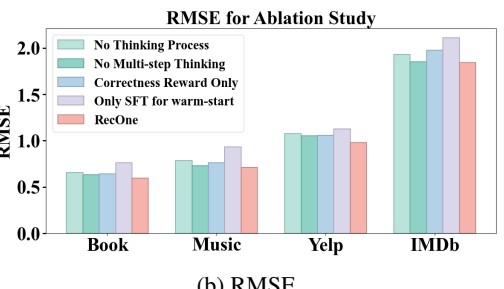

(a) MAE                                  (b) RMSE

Figure 4: 4a and 4b separately show the MAE and RMSE performance of RecOne and its four ablation variants on four datasets.

The experimental results are illustrated in Figure 4. Our RecOne model demonstrates significant superiority over alternative approaches in both MAE and RMSE metrics. Through experimental observations, we can discern that the **No Thinking Process** baseline underperforms the **No Multi-step Thinking** version in both metrics, which in turn is inferior to the complete RecOne model employing the three-step thinking process (Extract User Interest, Summarize Item Key Aspects, Evaluate Compatibility). This robustly validates the positive impact of our proposed strategy of explicitly guiding the model through multi-step thinking on rating prediction accuracy. Furthermore, the alternative version **Correctness Reward Only** exhibits a notable decline in both MAE and RMSE metrics compared to RecOne. This phenomenon can be attributed to the fact that the correctness-only reward mechanism encourages the model to generate integer ratings that exactly match user scores, rather than decimal values approximating the actual ratings. Such mechanism suppresses the model's tendency to produce predictions close to actual ratings, instead promoting aggressive predictions that completely align with target scores. Finally, the contrast model employing **Only SFT for warm-up** demonstrates the poorest performance among all baseline methods. This strongly substantiates the significance of our proposed approach of collaboratively optimizing the reasoning process and rating prediction task through RL, acquiring reasoning ability through interaction and optimization, rather than merely obtaining surface-level reasoning through SFT for enhancing the model's predictive accuracy.

## 4.4 Cost–Effectiveness and Practical Deployment (RQ4)

We follow exactly the same experimental protocol as in the previous sections and, on the Amazon-Music dataset, additionally evaluate two further variants:

- **RecZero(early–stop)** — training stops once it attains the same MAE as the external best SFT baseline;
- **RecOne(SFT)** — the RL phase is removed and the model is trained with SFT until convergence.

Table 2 reveal four main findings:

- **Larger performance gain per unit cost.** The pure-SFT variant RecOne(SFT) consumes 20 K labelled instances, $\sim$0.6 GPU-hours, and 597 inference tokens per request to reach an MAE of 0.6472. By contrast, RecZero(early–stop) already surpasses this score (MAE = 0.5419) while using only 0.48 K labels ($-97.6\%$), 0.4 GPU-hours ($-33\%$), and significantly smaller serving budget (331 tokens). Extending RL to full convergence (RecZero) further pushes the error down to MAE/RMSE = 0.4271/0.7058, achieving a 21.5% MAE reduction against the strongest pure-SFT baseline (Reason4Rec) with merely 12% of its labels—yielding more than a $5\times$ gain in *error reduction per 1K labels*.

- **Fast adaptation to cold-start and distribution drift.** Commercial recommenders face a constant influx of new items and shifting user interests. RecZero can be re-optimized with merely a few hundred fresh on-line interactions collected within minutes, whereas SFT pipelines must accumulate and curate a much larger batch before retraining.

- **The simplest pipeline among reasoning-enhanced methods.** RecZero keeps (1) a single model, (2) a single training stage, (3) no distillation teacher and (4) end-to-end optimization, resulting in markedly lower engineering overhead than multi-stage alternatives such as EXP3RT or Reason4Rec, which each juggle three models and at least two training phases.

Table 2: Training and inference cost of different reasoning–enhanced recommenders

| Method | Paradigm | Samples | Models | Training Time | Avg.Inf.Tokens | Inf.Stages | Inf.Token |
|--------|----------|---------|--------|---------------|----------------|------------|-----------|
| EXP3RT | SFT | 20K | 3 | 1.2h | 187.63 | 3 | 562.89 |
| Reason4Rec | SFT | 20K | 3 | 1h | 175.49 | 2 | 350.98 |
| RecZero(early-stop) | RL | 0.48K | 1 | 0.4h | 331.64 | 1 | 331.64 |
| RecOne(SFT) | SFT | 20K | 1 | 0.6h | 597.32 | 1 | 597.32 |
| RecZero | RL | 2.4K | 1 | 1.1h | 310.52 | 1 | 310.52 |
| RecOne | SFT+RL | 2.6K | 2 | 1.4h | 412.79 | 1 | 412.79 |

- **RL complements rather than replaces SFT.** RecOne(full) shows that starting from an SFT warm-start and applying a short RL refinement can push the error down to MAE/RMSE = 0.3816/0.6776. In practice, one may first perform a quick SFT pass on large historical logs, then run a lightweight RecOne-style RL fine-tuning on the latest or strategically important traffic.

In summary, the experiments reveal four clear takeaways. First, pure SFT is outperformed by pure reinforcement learning: despite requiring 20 K labelled samples, longer training time and almost twice the inference cost, RecOne(SFT) still lags behind the early-stopped RL variant, RecZero(early-stop). Second, when allowed to converge fully, RecZero further cuts MAE by 21.5 % relative to the strongest SFT baseline, without demanding additional computational resources. Third, both RecZero and RecOne can be re-optimized using only a few hundred fresh interactions, making them well suited to sparse-data or drifting-distribution scenarios. Finally, both methods preserve a single-model, teacher-free pipeline, offering a markedly simpler and more cost-efficient solution than existing multi-stage reasoning-enhanced recommenders.

# 5 Related Work

## 5.1 LLM-based Explainable Recommendation

The remarkable capabilities demonstrated by LLMs in world knowledge understanding and contextual processing have inspired researchers to leverage LLMs for handling rich semantic information to provide recommendation suggestions [41–45]. DRDT [46] employs a prompt-based approach that requires LLMs to analyze user preferences from multiple dimensions, enhancing sequential recommendations through critic prompts for reflection. GOT4Rec [23] encourages LLMs to engage in multi-perspective thinking for recommendations using the graph of thought strategy. However, these methods primarily adopt few-shot approaches for recommendation tasks, constrained by the inherent limitations of LLMs in the recommendation domain.

Recent studies have explored fine-tuning LLMs to optimize their reasoning capabilities for recommendation. Rec-SAVER [20] utilizes larger-scale LLMs to generate rationalized intermediate reasoning, fine-tuning smaller models to enhance their recommendation reasoning abilities. EXP3RT and Reason4Rec [22, 21] decompose the reasoning process into multiple steps for independent training, distilling the reasoning capabilities of larger models into smaller ones, and separately optimizing rating tasks for better performance. While these efforts have significantly enhanced the accuracy of LLMs in performing reasoning-enhanced rating prediction tasks, challenges such as suboptimal reasoning trajectories and decoupled training objectives remain unresolved.

## 5.2 LLM Post-training

As the most widely-used post-training technique, SFT quickly adapts a pretrained LLM to downstream tasks, yet its impact on model generalization remains a matter of debate. Recent studies have investigated how SFT influences the generalization of foundation models. SFT Memorizes, RL Generalizes [30] shows that SFT tends to memorize training data and thus generalizes poorly to out-of-distribution scenarios, while RL strengthens a model's core abilities and enhances domain generalization. SFT or RL [29] systematically compares SFT and RL and reveals that SFT often leads to the imitation of "pseudo reasoning paths" generated by an expert model. These paths may look similar to the native reasoning produced by RL-trained models, yet they usually contain redundant, hesitant, or low-information steps and sometimes even incorrect reasoning.

RL [47, 48] operates through the interaction mechanism between agents and the environment, rewarding correct actions with the core objective of maximizing cumulative returns. In recent years,

researchers have introduced RL into the fine-tuning process of LLMs. Among these approaches, Reinforcement Learning with Human Feedback (RLHF) [49] typically employs the Proximal Policy Optimization (PPO) [50] algorithm to achieve alignment with human preferences. However, the PPO algorithm faces technical challenges related to excessive memory consumption. To simplify the application of RL in LLMs, researchers have proposed innovative methods such as Direct Preference Optimization (DPO) [51]. In the context of recommendation systems, the s-DPO method [52] combines DPO with the softmax loss function, enhancing LLMs' capability in mining hard negative samples for recommendation systems. Although these methods have improved efficiency, they still face challenges such as off-policy issues and performance ceilings that fall short of online methods. The recently proposed Group Relative Policy Optimization (GRPO) [31] further reduces memory requirements by estimating group scores, while replacing traditional reward models with rule-based reward mechanisms. Through this innovative paradigm, LLMs can significantly enhance their reasoning capabilities and domain-specific performance (e.g., in mathematics and programming) without requiring annotated data [32, 33]. Recent studies [53–55] have further explored the applications of RL in mathematics and general domains. Despite these advancements, the application of RL in improving rating prediction performance through reasoning capabilities remains unexplored.

## 6    Conclusion

In this work, we introduced RecZero, a novel RL paradigm for LLM-based recommendation systems, addressing key limitations of conventional distillation-based methods. By unifying reasoning and recommendation into a single LLM trained through structured prompts and rule-based rewards, RecZero eliminates the need for separate teacher-student models and enables continuous optimization of reasoning processes. Our experiments across multiple benchmarks demonstrate RecZero's superiority over existing approaches, highlighting its ability to develop autonomous reasoning capabilities aligned with recommendation objectives.

Additionally, we explored RecOne, a hybrid paradigm combining supervised fine-tuning with RL, which further elevates the performance ceiling of the model in RL. Both approaches showcase the potential of RL in advancing recommendation systems, paving the way for more adaptive and efficient models.

### Acknowledgments and Disclosure of Funding

This research is supported by the Fundamental Research Funds for the Central Universities (WK2100250065) and the National Natural Science Foundation of China (62302321).

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

Figure 5: System Prompt

# A  System Prompt

As illustrated in Fig 5, we guide the early outputs of the LLM through a structured process in the system prompt to achieve faster training convergence and superior model performance. Specifically, within the `<analyze user>` and `</analyze user>` tags, we directed the LLM to first list the features [like] and disliked features [dislike] of each product based on the user's historical interaction records, and then summarize the user's complete preferences using [pos] and [neg] tags. Subsequently, for the target item, we encourage the LLM to use [like] and [dislike] tags within the `<analyze item>` and `</analyze item>` tags to summarize the features that the user might like and dislike about the target item. Following this, the LLM engaged in a thoughtful analysis of the match between the user and the target item within the tags `<match></match>`, and finally provided the predicted user rating within the tags `<rate></rate>`.

# B  Limitation

While RecZero and RecOne exhibit promising results, this study is not without its limitations. First, due to computational constraints, we were unable to fully assess the potential performance gains that could be achieved by leveraging larger base models as the foundation for RL. This limitation may hinder a thorough evaluation of the models' true performance potential. Second, the study did not explore whether RecZero and RecOne could serve as viable replacements for existing Teacher models in generating cold-start data for multi-round self-iterative optimization. These limitations highlight the need for future work to investigate the scalability and efficacy of RecZero and RecOne in more resource-intensive and complex iterative settings, thereby providing a more comprehensive assessment of their practical applicability.

# C  Experimental Design and Evaluation

### Datasets

To comprehensively validate the effectiveness of our proposed RecZero and RecOne, we conduct systematic experiments on four representative real-world recommendation datasets. Through comparative analysis with various baseline models, including traditional recommendation methods and state-of-the-art (SOTA) LLM-based approaches, we thoroughly demonstrate the superiority of our proposed methods. These datasets span across different domains and scenarios, specifically including: Book, Music and Yelp.

- **Book:**
  This dataset is derived from the "Book" subset of the widely used Amazon dataset for recommendation scenario evaluation. It records a large number of ratings, reviews, and rich book product metadata from users on the Amazon platform in the book scenario.

- **Music:**
  Similarly, this dataset is from the "Music" subset of the widely used Amazon dataset for recommendation scenario evaluation, containing user ratings and reviews in the music domain.

- **Yelp:**
  The Yelp Open dataset contains a large number of ratings and reviews from consumers for local restaurants and stores, and is widely used for performance evaluation in recommendation scenarios.

**Baselines.**

We compare RecZero and RecOne with a broad set of baselines that cover traditional collaborative filtering (CF) methods, review-based methods, and LLM-based approaches: MF, which is a classic collaborative filtering method; DeepCoNN, NARRE, and DAML, which rely on review text to learn richer user and item representations. DeepCoNN uses CNNs for joint modeling, NARRE applies attention to select informative reviews, and DAML models interactions between users and items. We also include Rec-SAVER, EXP3RT, and Reason4Rec, which are based on large language models. Rec-SAVER takes users' historical interactions and target item metadata as input to a teacher model to generate intermediate reasoning traces, which are distilled into a smaller student model to strengthen rating prediction. EXP3RT and Reason4Rec decompose a single step prediction into multiple reasoning steps, improving accuracy on reasoning enhanced rating prediction.

**Training Protocol.**

In our study, we employ the Qwen2.5-7B-Instruct-1M model as the starting point for RL due to its strong instruction-following capabilities and planning abilities acquired during pre-training. For traditional baseline experiments, we utilize a single H20 GPU, while the LLM-based baselines and our RecZero and RecOne frameworks are executed on an 8-card H20 GPU setup. All experiments are conducted using Python 3.9.

**Evaluation Protocol.**

For the rating prediction task, we employ Mean Absolute Error (MAE) and Root Mean Square Error (RMSE) as evaluation metrics. MAE measures the prediction accuracy of the model by calculating the average of the absolute errors between predicted values and true values, where a smaller value indicates better prediction performance. RMSE evaluates the model's predictive capability by computing the square root of the average of the squared errors between predicted values and true values. It is more sensitive to larger errors and provides a better reflection of the overall prediction stability of the model.

# D   Statistics

For both RecZero and RecOne, we utilize the Qwen2.5-7B-Instruct-1M model as the starting point for RL. During training, the batch size is set to 8, and the learning rate is 2e-6. Each data sample undergoes 8 rollouts during the training process. We set the sampling temperature to 1.0, the training epoch to 1, and the KL divergence to 0. Additionally, for the cold-start process of RecOne, we employ reasoning data provided by DeepSeek-R1 for cold-start initialization. We partition the dataset and select 1000 data samples that are not included in either the training or test sets for the cold-start experiments.

