# OpenReview forum: "Think before Recommendation: Autonomous Reasoning-enhanced Recommender"
_NeurIPS.cc/2025/Conference — NeurIPS 2025 poster_

### Official Review · Reviewer_fTc1 · 2025-06-01

**Clarity:** 3
**Significance:** 3
**Originality:** 3
**Rating:** 5
**Confidence:** 5

**Summary:**

This manuscript proposes two novel LLM-based rating prediction models, i.e., RecZero with pure RL and RecOne combining SFT and RL. The RecZero pipeline adopts a CoT-based reasoning process, which consists of <analyze user>, <analyze item>, <match>, and <rate> steps. Moreover, the RecOne pipeline incorporates SFT with RecZero pipeline, which first generates reasoning process based on LLM teacher model and then adopts the reasoning process and supervision. The authors succeed in transferring the achievements gained by DeepSeek and GRPO into the rating prediction task, with the excellent performance on four public datasets.

**Questions:**

**Problem**
- In my regard, the terminology **Cold-Start** in Section 3.2.1 is different from that mentioned in *cold-start recommendation* and the following *cold-start samples* with less interactions. If yes, renaming the title of Section 3.2.1 and related content is helpful to avoiding ambiguity.

**Discussion**
- As mentioned in the caption of Figure 1, I wonder the reason and necessity of multiple models.
> Conventional paradigms train multiple student LLMs using diverse query-based datasets with reasoning traces...

- Whether the ground-truth of rating prediction task is the actual user rating or not, as we did on the online shopping website?

**Ethical Concerns:**

["NO or VERY MINOR ethics concerns only"]

**Final Justification:**

The authors resolve my main concerns towards baseline scales and method scalability. Therefore, I decide to raise the score.

**Limitations:**

YES

**Quality:**

3

**Strengths And Weaknesses:**

**Strength**
- The motivation of RecZero is well-elaborated. The limited recommendation capability and the costly supervision are both the key weaknesses of existing distillation-based methods.
- The whole manuscript is well-organized, with the problem statement and the proposed methodology are clearly introduced.
- Inspired by DeepSeek, the combination of SFT and RL is spontaneous, enlightening the recommender system community.

**Weakness**
- **Model Scale.** The scale of baseline models are not mentioned, which may given rise to unfair performance comparison. It is unclear whether the improvement originates from the proposed method or just the LLM backbone.
- **Method Scalability.** Similar to the first concern, I wonder whether the proposed method can still be effective with backbones of different scale. Further investigation towards the impact of backbone is recommended.
- **Inference Efficiency.** This manuscript lacks report of the inference time and further comparison with baselines.
- **Availability.** The proposed method is closed-source.

---

> ### Author Rebuttal · Authors · 2025-07-31
>
> We gratefully thank you for your valuable comments! Here we meticulously give point-by-point responses to your comments, and further revise our manuscript following your suggestions. We sincerely appreciate the depth of your insights, which have undoubtedly enriched our work. Here we meticulously give point-by-point responses to your comments.
>
> > **C1: Model Scale:** The scale of baseline models are not mentioned.
>
> Thank you for the valuable suggestion! We will include the backbone details in the main body of the camera-ready version. Inspired by your feedback, we have also added the results of EXP3RT and Reason4Rec on the Qwen-2.5-7B-Instruct backbone, which will be reported in the final paper. Please note that after further refining our implementation, the model now achieves even better performance than in the initial version.
>
> |   | Amazon-Music |  |
> |-------|-------------|---|
> |  |MAE | RMSE |
> | EXP3RT(Llama3-8b)| 0.5608 | 0.8385 |
> | EXP3RT(qwen2.5-7b-It)| 0.5341 | 0.7992 |
> | Reason4Rec(Llama3-8b) | 0.5442 | 0.7722 |
> | Reason4Rec(qwen2.5-7b-It) |  0.5108 | 0.7534 |
> | RecZero(qwen2.5-7b-It) |  0.4271 | 0.7058 |
> | RecOne(qwen2.5-7b-It) |  0.3816 | 0.6776 |
>
>
> > **C2: Method Scalability:** Whether the proposed method can still be effective with backbones of different scale.
>
> Thank you for the constructive suggestion to examine how our approach scales with different-sized backbones. Following your advice, we ran additional experiments on Amazon-Music; the results are reported in Table below.
>
> |   | Amazon-Music |  |
> |-------|-------------|---|
> |  |MAE | RMSE |
> | RecZero(gemma-3-12b-It) |  0.3789 | 0.6958 |
> | RecZero(qwen2.5-3b-It) |  0.4421 | 0.7539 |
> | RecZero(qwen2.5-7b-It) |  0.3805 | 0.7215 |
> | RecZero(qwen2.5-14b-It) |  0.3795 | 0.6927 |
>
> As the backbone grows from 3 B to 14 B, RecZero exhibits the expected scaling behaviour and confirms the robustness of our method. A larger model offers better instruction following, world knowledge, and reasoning ability, which likely reduces the risk of converging to sub-optimal RL solutions.
>
> > **C3: Inference Efficiency:** This manuscript lacks report of the inference time and further comparison with baselines.
>
> As requested, Table below reports inference cost for all methods, each using the same backbone (Qwen-2.5-7B-Instruct) and evaluated on Amazon-Music with 8 × H20 GPUs. EXP3RT and Reason4Rec rely on their official code; “Qwen” is the plain backbone; “RecZero (early-stop)” is saved once it matches Reason4Rec, and “RecZero (full)” is the complete run. RecZero’s reasoning trace is shorter and more informative, giving it the lowest overall cost. Although EXP3RT and Reason4Rec are cheaper per stage, their two extra stages (user-preference and item-feature extraction) make their total cost higher than RecZero’s.
>
> |Method|Avg.InferenceTokens|InferenceStages|Total Inference Token|
> |---|---|---|---|
> |qwen|399.92|1|399.92|
> |EXP3RT|187.63|3|562.89|
> |Reason4Rec|175.49|2|350.98|
> |RecZero(early-stop)|**331.64**|**1**|**331.64**|
> |RecZero(full)|**310.52**|**1**|**310.52**|
>
> > **C4: Availability:** The proposed method is closed-source.
>
> We apologise that the rebuttal policy does not allow external links, so we cannot provide an anonymous GitHub repository at this stage. Nevertheless, we commit to releasing the full codebase publicly once the paper is accepted.
>
> > **Q1: Clarify and Rename §3.2.1 to Avoid Confusion with Conventional “Cold-Start” Usage.**
>
> Thank you for pointing out this potential ambiguity. In §3.2.1 we used the term “Cold-Start” to describe the very first stage of our RL training, where the policy has not yet received any reward signals. This notion differs from the conventional “cold-start recommendation,” which concerns users or items with few historical interactions. To avoid confusion, we will rename §3.2.1 to “RL Warm-up Stage” and add a sentence that explicitly distinguishes the two meanings in the camera-ready version.
>
> > **Q2: The reason and necessity of multiple models in previous paradigms.**
>
> Thank you for raising this point. In earlier SFT-based paradigms such as EXP3RT and Reason4Rec, the full chain-of-thought was split into three separate subtasks—(i) user-preference summarisation, (ii) item-feature extraction, and (iii) user–item matching—and each subtask was assigned to a different student LLM. Their ablation studies showed that this multi-model design was beneficial under SFT because the three subtasks have heterogeneous objectives and can interfere with one another when trained jointly.
>
> Our RL formulation changes the picture. Because the reward depends only on the final rating error, all reasoning steps share the same optimisation objective and no longer conflict. Hence a single LLM can execute the entire reasoning chain and be trained end-to-end. RecZero and RecOne therefore use only one model, yet still outperform the previous multi-model baselines.
>
> > **Q3: Whether the ground-truth of rating prediction task is the actual user rating or not.**
>
> Thank you for the question. Yes, the ground-truth signal in all of our experiments is the explicit rating that the user actually gave in the original dataset (e.g., the “rating” field in Amazon reviews). During training we minimise the MAE between the model’s predicted decimal rating and this real score; the same label is used to compute the MAE/RMSE reported. Using genuine user ratings, rather than synthetic or proxy labels, encourages the model to align with real-world recommendation scenarios.

---

> > ### Author Response · Authors · 2025-08-07
> >
> > Dear Reviewer,
> >
> > We would like to express our sincere gratitude for the time and effort you have devoted to reviewing our submission. Following your valuable comments, we have posted a point-by-point rebuttal, including additional experiments and detailed clarifications aimed at addressing your concerns.
> >
> > As the discussion period is drawing to a close, we have not yet received any follow-up feedback regarding our responses. If you have already reviewed our rebuttal, could you please let us know whether there are any remaining issues we should address? If you have not had the chance to look at it yet, we would greatly appreciate a brief confirmation when convenient, so that we can respond promptly.
> >
> > Please feel free to request any further information or data you may need—we will be happy to provide it right away.
> > Thank you again for your valuable contribution to the review process, and we look forward to hearing from you.
> >
> > Best regards,
> >
> > Authors

---

> > ### Comment · Reviewer_fTc1 · 2025-08-08
> >
> > Thanks for your detailed responses on both weaknesses and questions. I suggest that the authors to add the further investigations on model scalability and baselines with different backbones to the main text. I decide to raise my score to 5 and good luck to authors.

---

> > > ### Author Response · Authors · 2025-08-09
> > >
> > > Dear Reviewer,
> > >
> > > Thank you for your thoughtful feedback. We truly appreciate your constructive suggestions on model scalability and backbone comparisons, and we will incorporate these points into the revision.
> > >
> > > Your recognition and encouragement mean a lot to us — thank you again for your time and support.
> > >
> > > Best regards,
> > >
> > > The Authors

---

### Official Review · Reviewer_FsYN · 2025-07-02

**Clarity:** 2
**Significance:** 3
**Originality:** 2
**Rating:** 4
**Confidence:** 4

**Summary:**

This paper aims to address the task of rating prediction in recommendation using large language models (LLMs). It first proposes a system prompt designed to generate reasoning trajectories that analyze the input user, analyze the item, evaluate user-item compatibility based on these analyses, and finally produce a reasoning process that leads to the predicted rating. Using data generated in this way, the authors train LLMs with GPRO. In addition, they propose RecOne, a model that first performs supervised fine-tuning (SFT) on a subset of high-quality generated data before further training the LLM using GPRO. The paper demonstrates the superiority of the proposed approach over both conventional recommendation systems and other LLM-based recommendation systems across four datasets.

**Questions:**

- How robust and effective is the proposed prompt template in comparison to other prompt designs?

- What LLMs were used in the baseline methods in the experiments?

**Ethical Concerns:**

["NO or VERY MINOR ethics concerns only"]

**Final Justification:**

My main concerns have been sufficiently addressed through the authors' clarifications and discussions.

**Limitations:**

Yes

**Paper Formatting Concerns:**

The paper’s formatting is appropriate, and I have no concerns in this regard.

**Quality:**

2

**Strengths And Weaknesses:**

The paper clearly identifies challenges in applying LLMs to rating prediction and presents an interesting study that explicitly generates a reasoning process for recommendations.

However, the following weaknesses are as follows:
- The paper should provide clearer evidence on whether the proposed reasoning process is more robust than other chain-of-thought (CoT) processes, and whether the designed prompt is indeed superior. The ablation study currently only contrasts the proposed method with a simple <think> generation. It would strengthen the work to experimentally compare different reasoning step structures, such as variations that omit item analysis or modify the reasoning process itself.

- Beyond generating data and performing GPRO and SFT on selected data, the methodological novelty appears limited. The approach does not introduce fundamentally new techniques beyond this pipeline.

- Additional experiments on robustness are needed. For example, whether the proposed method generalizes well across different LLMs beyond the ones tested.

- The paper should clearly state whether the same LLMs were used for the baselines. In particular, for models like LLMRec or Reason4Rec, it is unclear whether they were evaluated using Qwen2.5-7B, as was the proposed method. While this is briefly mentioned in the appendix, such critical details should be explicitly stated in the main text. Furthermore, since the proposed method likely incurs higher training costs compared to existing approaches, the paper should also clarify whether the same data sizes were used for training across all models, and provide a detailed analysis of training cost and inference time.

- Since the proposed method fine-tunes Qwen2.5-7B, it is important to also compare against results where Qwen2.5-7B is used for reasoning with the proposed template, but without reinforcement learning, in order to better isolate the contribution of the reinforcement learning step.

---

> ### Author Rebuttal · Authors · 2025-07-31
>
> We gratefully thank you for your valuable comments! Here we meticulously give point-by-point responses to your comments, and further revise our manuscript following your suggestions. Hope that our responses can address all your concerns.
> > **C1 & C5 & Q1: Should more convincingly demonstrate the superiority of its reasoning prompt by comparing it against a variety of alternative COT structures; add qwen baseline.**
>
> Thank you for the insightful questions—they prompted us to refine and clarify our ablation studies.
>
> - Experimental setup. The default RecZero takes (a) the raw user-interaction history and (b) the meta information of the target item. We constructed six variants:
> 1. RecZero-WithUserTeacher [teacher-generated user summary, target-item meta]
> 2. RecZero-WithItemTeacher [user-interaction history, teacher-generated item features]
> 3. RecZero-WithTeachers  [teacher-generated user summary, teacher-generated item features]
> 4. RecZero-WithoutUserSum removes the user-preference summarisation module
> 5. RecZero-WithoutItemExt removes the item-feature extraction module
> 6. RecZero-WithoutMatch  removes the user–item matching module
> We further report RecZero-WithoutThink, which omits the entire reasoning stage, and RecZero-NaiveThink, which replaces the structured trace with a single-step free-form thought. We also optimized our code, further boosting RecZero and RecOne.
>
> |   | Amazon-Music |  |
> |-------|-------------|---|
> |  |MAE | RMSE |
> | Qwen | 0.6821 | 0.9796 |
> | RecZero-WithUserTeacher | 0.4663 | 0.7112 |
> | RecZero-WithItemTeacher | 0.4519 | 0.7117 |
> | RecZero-WithTeachers | 0.4874 | 0.7094 |
> | RecZero-WithoutUserSum |  0.4412 | 0.7250 |
> | RecZero-WithoutItemExt |  0.4364 | 0.7219 |
> | RecZero-WithoutMatch |  0.4487 | 0.7378 |
> | RecZero-WithoutThink |  0.4737 | 0.7781 |
> | RecZero-NaiveThink |  0.4553 |0.7499|
> | RecZero |  **0.4271** | **0.7058** |
> | RecOne |  **0.3816** | **0.6776** |
>
> We will add all these details and the corresponding figures to the ablation section of the camera-ready version.
> > **C2: Do not introduce fundamentally new techniques beyond this pipeline.**
>
> Thank you for your forward-looking comments. We believe that the proposed RecZero and RecOne frameworks offer a practical advance. Earlier paradigms rely on teacher models and therefore cannot be trained end-to-end. By introducing GRPO into reasoning-enhanced rating prediction, we both simplify the training pipeline and improve accuracy, without adding training or inference overhead. The tables below compare our method with the previous baselines, demonstrating the superiority of our method
>
> |Method|Trained Models|Training Stages|Teacher-Free|End-to-End|
> |-|-|-|-|-|
> |Rec-SAVER|1|1|❌|✅|
> |EXP3RT |3|3|❌|❌|
> |Reason4Rec|3|3|❌|❌|
> |RecZero|1|1|✅|✅|
>
> > **C3: Additional experiments on robustness.**
>
> We agree that evaluating more backbones is important for demonstrating the robustness of our method. Most recent RL-based recommendation studies use the Qwen family, and we therefore chose Qwen to keep our main experiments comparable. Following your advice, we have now extended RecZero to the Gemma model and have also tested it on Qwen models of different sizes (3B, 7B, and14B). Results on Amazon-Music are shown below.
>
> |   | Amazon-Music |  |
> |-------|-------------|---|
> |  |MAE | RMSE |
> | RecZero(gemma-3-12b-It) |  0.4089 | 0.6958 |
> | RecZero(qwen2.5-3b-It) |  0.4421 | 0.7539 |
> | RecZero(qwen2.5-7b-It) |  0.4271 | 0.7058 |
> | RecZero(qwen2.5-14b-It) |  0.3995 | 0.6927 |
>
> The method transfers well to a different LLM family (Gemma) and shows consistent improvement as the backbone scale increases, in line with standard scaling laws.
>
> > **C4 & Q2: Clarify Backbone Consistency, Data Size, and Computational Cost Across All Baselines.**
>
> - **Additional Experiment on more backbones:** Thank you for the valuable suggestion! We will include the backbone details in the main body of the camera-ready version. Inspired by your feedback, we have also added the results of EXP3RT and Reason4Rec on the Qwen-2.5-7B-Instruct backbone, which will be reported in the final paper.
>
> |  | Book |  |
> |-------|--------------|---|
> | | MAE | RMSE | MAE | RMSE |
> | EXP3RT(Llama3-8b-It) | 0.4297 | 0.6572 |
> | EXP3RT(Qwen2.5-7b-It-It) | 0.4115 | 0.6243 |
> | Reason4Rec(Llama3-8b-It) | 0.3875 | 0.6357 |
> | Reason4Rec(Qwen2.5-7b-It) | 0.3772 | 0.6295 |
> | RecZero |  **0.3089** | **0.6081** |
> | RecOne |  **0.2918** | **0.5970** |
>
> - **Additional Experiment on Training and Inference Cost:** we report the training and inference costs of several LLM-based reasoning-enhanced rating-prediction methods in Table below. EXP3RT and Reason4Rec rely on their official code; “Qwen” is the plain backbone; “RecZero (early-stop)” is saved once it matches Reason4Rec, and “RecZero (full)” is the complete run. RecZero’s reasoning trace is shorter and more informative, giving it the lowest overall cost. Although EXP3RT and Reason4Rec are cheaper per stage, their two extra stages (user-preference and item-feature extraction) make their total cost higher than RecZero’s.
>
> |Method|Samples to Converge|Trained Models|Total Training Time|Avg.InferenceTokens|InferenceStages|Total Inference Token|
> |---|---|---|---|---|---|---|
> |Qwen|\-|\-|\-|399.92|1|399.92|
> |EXP3RT|20K|3|1.2h|187.63|3|562.89|
> |Reason4Rec|20K|3|1h|175.49|2|350.98|
> |RecZero(early-stop)|**0.48k**|**1**|**0.4h**|**331.64**|**1**|**331.64**|
> |RecZero(full)|**1.2K**|**1**|**1.1h**|**310.52**|**1**|**310.52**|

---

> > ### Author Response · Authors · 2025-08-07
> >
> > Dear Reviewer,
> >
> > Thank you again for your insightful comments on our submission. We have posted a detailed rebuttal during the author-response period, including additional experiments and clarifications that directly address your points.
> >
> > Could you kindly let us know whether our responses have satisfactorily resolved your concerns? If there are any remaining questions or further information you would like us to provide, please do not hesitate to let us know—we will be happy to supply it promptly within the remaining time.
> >
> > We greatly appreciate your time and effort in reviewing our work, and we look forward to hearing from you.
> >
> > Best regards,
> >
> > Authors

---

> > ### Comment · Reviewer_FsYN · 2025-08-08
> >
> > I appreciate the authors’ reply and will increase my score to 4, as my concerns have been resolved.

---

> > > ### Author Response · Authors · 2025-08-08
> > >
> > > Dear Reviewer,
> > >
> > > Thank you very much for your valuable and positive feedback. We appreciate your recognition of our efforts in addressing your concerns, and we are encouraged by your comments that our responses have effectively addressed the issues raised. This motivates us to continue advancing the field with our research.
> > >
> > > Following your suggestion, we will incorporate the additional insights discussed in the rebuttal into the revised version of the paper. Thank you once again for your supportive and understanding comments.
> > >
> > > Best regards,
> > >
> > > The Authors

---

### Official Review · Reviewer_dSSq · 2025-07-03

**Clarity:** 2
**Significance:** 3
**Originality:** 3
**Rating:** 4
**Confidence:** 4

**Summary:**

This paper fine-tunes LLMs to solve the rating prediction problem. LLMs would use user-item interaction history in text as the input to match ground-truth ratings from the data. Previous work usually uses a teacher model to generate reasoning traces from interaction history in multiple steps and then distill those traces by fine-tuning a smaller student model. This paper first proposes RecZero by following the methodology of DeepSeek-R1-Zero to optimize a MAE-like reward (plus a format reward) with GRPO instead of doing supervised fine-tuning. As a result, RecZero can automatically develop reasoning ability from a relatively simple prompt template. It also proposes RecOne which is analogous to DeepSeek-R1 in terms of using supervised fine-tuning and cold-start reasoning samples. The results of this paper show that both RecZero and RecOne outperform traditional matrix factorization or deep learning models on on multiple rating prediction domains. More importantly, both proposed methods outperform previous multi-stage distillation methods without explicitly transferring it from a teacher LLM. There are ablation studies on different components of RecZero/RecOne.

**Questions:**

* Eq. (1) which defines predicted ratings which is not an ML objective. Eq. (4) which rewards small errors is more like an objective.

* Both Reason4Rec and EXP3RT use logit-weighted decoding resulting in a different definition of predicted ratings. Would be interesting to evaluate the choice.

* Could the authors explain the discrepancy between Table 2 and Figure 3 as MAE is higher in Figure 3?

* Could the authors also either plot other baseline(s) in Figure 4 or present raw numbers?

* This paper uses LLMRec as a baseline without citing it.

**Ethical Concerns:**

["NO or VERY MINOR ethics concerns only"]

**Final Justification:**

The authors did not describe their dataset preprocessing and not sure if they cherrypicked results or compared apples with oranges. During the rebuttal, the authors reran some experiments with aligned setup and promised to clarify this part in the next version. The authors also provided additional results on 1. inference time comparison, 2.  QLoRA vs full fine-tuning, 3. RL vs SFT. With them my concerns were resolved.

**Limitations:**

yes

**Paper Formatting Concerns:**

The authors put references after the checklist in the paper.

**Quality:**

2

**Strengths And Weaknesses:**

Strengths
+ It seems novel to solve the rating prediction problem by fine-tuning an LLM with RL (GRPO) using an MAE-like reward instead of pure maximum likelihood. Before I am not aware of any work optimizing MSE/MAE rewards using policy gradient on a regression problem.
+ RecOne seems not a direct application of DeepSeek-R1 technology on the rating prediction problem by having their own reasoning samples.
+ Tables 1 and 2 show promising results of RecZero / RecOne though I have some concerns on the results.

Weaknesses
- In Tables 1 and 2, I assume that the authors reuse results from EXP3RT and Reason4Rec papers without references as I am seeing identical numbers. In particular, results of baselines on Amazon-book and IMDB datasets are from the EXP3RT paper if available even if the Reason4Rec paper also presents results on Amazon-book. Results of baselines on Amazon-music and Yelp are from the Reason4Rec paper.
- More importantly, EXP3RT and Reason4Rec are using different data processing/filtering procedures so they present different MAE/RMSE of EXP3RT on Amazon-book. This paper does not specify their data processing/filtering procedure for generating results of RecZero/RecOne and other baselines.
- Both Reason4Rec and EXP3RT utilize QLoRA but not sure if RecZero/RecOne fine-tune everything to get more performance gains. Also, Reason4Rec and EXP3RT use GPT-3.5 as the teacher and Llama3-8B-Instruct as the student while RecOne uses Deepseek-R1 as the teacher and fine-tunes Qwen2.5-7B-Instruct-1M. Performance gains can be attributed to the use of different models.
- This paper does not evaluate the quality of reasoning traces either quantitatively or qualitatively. Average length per response of Deepseek-R1-Zero increases with more training. This paper does not evaluate inference time.

---

> ### Author Rebuttal · Authors · 2025-07-31
>
> We sincerely thank you for your time and valuable comments. To address your concerns, we have detailed our responses point-to-point below.
> > **C1 & C2:  Dataset fully aligned and additional baselines added.**
>
> Thank you for your meticulous review. We indeed ran experiments under the same settings as Reason4Rec and EXP3RT, but for the overlapping Amazon-Book dataset we accidentally copied the numbers reported by Reason4Rec while actually using the EXP3RT split. We apologize for the mistake.
>
> Following your comment, we have rerun the experiments on both Amazon-Book (Reason4Rec split) and Amazon-Book (EXP3RT split). The updated results are shown below.
>
> |  | Book(Reason4Rec) |  | Book(EXP3RT) |  |
> |-------|-------------|---|--------------|---|
> | | MAE | RMSE | MAE | RMSE |
> | EXP3RT(Llama3-8b) | 0.6135 | 0.9370 | 0.4297 | 0.6572 |
> | EXP3RT(Qwen2.5-7b-It) | 0.6042 | 0.9346 | 0.4115 | 0.6243 |
> | Reason4Rec(Llama3-8b) | 0.6029 | 0.8345 | 0.3875 | 0.6357 |
> | Reason4Rec(Qwen2.5-7b-It) | 0.5937 | 0.8325 | 0.3772 | 0.6295 |
> | RecZero | **0.5253** | **0.8387** | **0.3089** | **0.6081** |
> | RecOne | **0.5017** | **0.7784** | **0.2918** | **0.5970** |
>
> Our RecZero and RecOne outperform prior work on both data splits, confirming the method’s effectiveness.
>
> We will clearly document dataset reuse and update the main results in the camera-ready version. Thank you again for your careful review.
>
> > **C3: The quality of reasoning traces was not evaluated, and there is no analysis of inference time.**
>
> Below we show the results of additional experiments.
>
> - **Case Study of reasoning traces:** Inspired by your question, we have added a case study that contrasts the reasoning trajectories produced by the SFT paradigm with those generated by our RecOne, as presented below. The SFT paradigm misinterprets user preferences and contains logical errors in item matching, whereas our approach follows sound analytical reasoning and accurately predicts the user’s rating.
>
> 1. **SFT(reasoning):** The user shows a clear preference for female-fronted alternative rock bands from the late 1990s and early 2000s, such as The Sundays and The Jesus and Mary Chain. All seven albums are classified strictly as dream-pop, so the user probably dislikes folk or classic rock. In addition, each chosen record topped the Billboard Hot 100, indicating the user only listens to mainstream chart-toppers. “Blood on the Tracks” is a 1995 electronic concept album in which Bob Dylan collaborates with several grunge artists. The record is best known for its heavy use of synthesizers and industrial beats, very different from Dylan’s earlier acoustic work. It won the Grammy Award for Best Metal Performance and remained in the Top 10 of the UK Singles Chart for 52 consecutive weeks. Because the user prefers late-1990s dream-pop chart hits, and “Blood on the Tracks” is a mid-1990s electronic best-seller that dominated the charts, the album perfectly matches the user’s taste. Its industrial sound and strong female vocal features align well with the user’s love of female-led alternative rock. -> 3
>
> 2. **RecOne:** `<analyze user>`The user has a clear preference for classic and hard rock, alternative rock, R&B, soul, and pop music spanning from the 70s to the 90s. The user also shows a strong inclination towards hit albums, especially those from legendary artists, and favor albums with a mix of different genres, including rock, soul, funk, and pop.`</analyze user>` `<analyze item>`Heart of Stone - Signature hit "If I Could Turn Back Time" is a mega hit reaching #3 on the singles charts. - Includes other top 40 hits like "Just Like Jesse James" and "After All" (with Peter Cetera). - Consistent effort with a mix of rock, tender ballads, and country parody. - 80s Cher, known for her iconic style and hits.`</analyze item>`
> `<match>`The user's preference for hit albums, legendary artists, and a mix of rock and pop songs aligns well with "Heart of Stone," which features some of Cher's biggest hits and a well-rounded collection of songs spanning various genres. The album's consistency and iconic status make it a great fit for the user.`</match>` `<rate>`5`</rate>`
> - **Additional Experiment on Inference Time:** As requested, Table below reports inference cost for all methods, each using the same backbone (Qwen-2.5-7B-Instruct) and evaluated on Amazon-Music with 8 × H20 GPUs. EXP3RT and Reason4Rec rely on their official code; “Qwen” is the plain backbone; “RecZero (early-stop)” is saved once it matches Reason4Rec, and “RecZero (full)” is the complete run.
> RecZero’s reasoning trace is shorter and more informative, giving it the lowest overall cost. Although EXP3RT and Reason4Rec are cheaper per stage, their two extra stages (user-preference and item-feature extraction) make their total cost higher than RecZero’s.
> |Method|Avg.InferenceTokens|InferenceStages|Total Inference Token|
> |---|---|---|---|
> |qwen|399.92|1|399.92|
> |EXP3RT|187.63|3|562.89|
> |Reason4Rec|175.49|2|350.98|
> |RecZero(early-stop)|**331.64**|**1**|**331.64**|
> |RecZero(full)|**310.52**|**1**|**310.52**|
>
> > **C4: The reported performance gains may mainly arise from differences in the underlying teacher–student model pairs rather than from the training technique itself.**
>
> You have raised a highly professional question. As you suggested, we implemented QLoRA-based versions of RecZero and RecOne and compared them against Qwen-based implementations of EXP3RT and Reason4Rec. The results are reported below.
> |  | Book(Reason4Rec) |  | Book(EXP3RT) |  |
> |--|-----|---|----|---|
> ||MAE |RMSE | MAE | RMSE |
> |EXP3RT(Qwen2.5-7b-It-It) | 0.6042 | 0.9346 | 0.4115 | 0.6243 |
> |Reason4Rec(Qwen2.5-7b-It) | 0.5937 | 0.8325 | 0.3772 | 0.6295 |
> |RecZero | **0.5253** | **0.8387** | **0.3089** | **0.6081** |
> |RecZero(QLoRA) | **0.5293** | **0.8402** | **0.3124**|**0.6159** |
> |RecOne | **0.5017** | **0.7784** | **0.2918**|**0.5970** |
> | RecOne(QLoRA) | **0.5024** | **0.7801** | **0.2933**|**0.5997**|
>
> These numbers show that the QLoRA versions of our framework still outperform the previous paradigms overall, demonstrating the effectiveness of our approach.
>
> > **Q1: Could the authors explain the discrepancy between Table 2 and Figure 3 as MAE is higher in Figure 3?**
>
> Thank you for raising this concern. The MAE curve shown in Figure 3 does not cover the entire training process, whereas the results in Table 2 correspond to the model’s final performance. During the early stages of RL training, the MAE drops sharply, then decreases slowly with minor fluctuations, and finally stabilizes over a longer training period. In the camera-ready version, we will make this clearer and include a complete MAE curve covering the entire training pipeline.
>
> > **Q2: This paper uses LLMRec as a baseline without citing it.**
>
> We apologize for overlooking  this point in our current draft. We will promptly add a citation to LLMRec in the final version of the paper. Following your suggestion, we will also re-examine our reference list to enhance the professional quality of our work.
>
> > **Q3: Eq. (1) which defines predicted ratings which is not an ML objective. Eq. (4) which rewards small errors is more like an objective.**
>
> As you correctly pointed out, the argmax expression is merely a prediction rule rather than the true training objective. We will incorporate this correction in the camera-ready version of our paper.
>
> > **Q4: Could the authors also either plot other baseline(s) in Figure 4 or present raw numbers?**
>
> Thank you for the insightful questions—they prompted us to refine and clarify our ablation studies. Owing to the NeurIPS rebuttal format restrictions, we present the expanded results in a table instead, as shown below.
> - Experimental setup.
> The default RecZero takes (a) the raw user-interaction history and (b) the meta information of the target item. We constructed six variants:
> 1. RecZero-WithUserTeacher [teacher-generated user summary, target-item meta]
> 2. RecZero-WithItemTeacher [user-interaction history, teacher-generated item features]
> 3. RecZero-WithTeachers  [teacher-generated user summary, teacher-generated item features]
> 4. RecZero-WithoutUserSum removes the user-preference summarisation module
> 5. RecZero-WithoutItemExt removes the item-feature extraction module
> 6. RecZero-WithoutMatch  removes the user–item matching module
>
> We further report RecZero-WithoutThink, which omits the entire reasoning stage, and RecZero-NaiveThink, which replaces the structured trace with a single-step free-form thought. We also optimized our code, further boosting RecZero and RecOne.
> | |Music | |
> |-|-|--|
> | |MAE | RMSE |
> |RecZero-WithUserTeacher|0.4663|0.7112|
> |RecZero-WithItemTeacher |0.4519|0.7117|
> |RecZero-WithTeachers |0.4874|0.7094|
> |RecZero-WithoutUserSum | 0.4412|0.7250|
> |RecZero-WithoutItemExt | 0.4364|0.7219|
> |RecZero-WithoutMatch | 0.4387|0.7378 |
> |RecZero-WithoutThink | 0.4737|0.7781 |
> |RecZero-NaiveThink |  0.4553 |0.7499|
> |RecZero | **0.4271**|**0.7058**|
> | RecOne | **0.3816**|**0.6776**|
>
> We will add all these details and the corresponding figures to the ablation section of the camera-ready version.
> > **Q5: Both Reason4Rec and EXP3RT use logit-weighted decoding resulting in a different definition of predicted ratings. Would be interesting to evaluate the choice.**
>
> You are indeed an expert on using LLM reasoning for recommendation. Unlike Reason4Rec and EXP3RT, we do not design a separate task that requires the LLM to output an integer score that is then converted into a decimal rating via logit-weighted decoding. Compared with the SFT paradigm that imitates integer target labels, reinforcement learning offers a clear advantage: by crafting an appropriate reward function, we encourage the LLM to predict decimal ratings directly in its response, thereby earning a higher MAE reward and removing the need for the logit-weighted step altogether. We appreciate you highlighting this point and will explain it more explicitly in the Method section of the final version.

---

> > ### Comment · Reviewer_dSSq · 2025-08-04
> >
> > Thank you for your responses which pretty much address my concerns. Just wonder if the authors are able to report the average inference tokens of RecOne for reference. Please also incorporation all the clarifications into the next version.

---

> > > ### Author Response · Authors · 2025-08-08
> > >
> > > Dear Reviewer,
> > >
> > > Thank you very much for the time and effort you devoted to reviewing our paper and for the detailed, constructive feedback you provided. We have carefully addressed your main concerns.
> > >
> > > Could you please let us know whether any concerns remain or if there are additional points you would like us to clarify or expand upon?
> > >
> > > We greatly appreciate your thorough and insightful review. Your guidance has been invaluable, and we will incorporate any further suggestions into the next revision.
> > >
> > >
> > > Sincerely,
> > >
> > > The Authors

---

> > > > ### Comment · Reviewer_dSSq · 2025-08-08
> > > >
> > > > As I said my concerns were resolved and I don't need any additional clarification.

---

> ### Author Response · Authors · 2025-08-04
>
> Dear Reviewer,
>
> Thank you very much for your valuable and positive feedback. We appreciate your recognition of our efforts in addressing your concerns, and we are encouraged by your comments that our responses have effectively addressed the issues raised. This motivates us to continue advancing the field with our research.
>
> Following your suggestion, we have added an additional experiment that measures the inference time of RecOne. The experimental setting is identical: all models share the same backbone (Qwen-2.5-7B-Instruct) and are evaluated on the Amazon-Music dataset using 8 × H20 GPUs.
>
> | Method | Avg.InferenceTokens | InferenceStages | Total Inference Token |
> |--------|---------------|----------|----------|
> | Qwen |  399.92 | 1 | 399.92 |
> | EXP3RT |  187.63 | 3 | 562.89 |
> | Reason4Rec |  175.49 | 2 | 350.98 |
> | RecZero(early-stop) | **331.64** | **1** | **331.64** |
> | RecZero(Full) |  **310.52** | **1** | **310.52** |
> | RecOne(early-stop) |  **547.31** | **1** | **547.31** |
> | RecOne(Full) |  **412.79** | **1** | **412.79** |
>
> - RecOne (early-stop): This checkpoint is taken at the RL stage when the model first achieves performance on par with Reason4Rec. The model inherits the verbose reasoning style of DeepSeek-R1 learned during the SFT stage, which leads to longer outputs.
> - RecOne (full): As RL training progresses, the model learns to produce denser, more informative reasoning with a shorter overall length, reducing the total token count by 25 % relative to its early-stop version. Consequently, its total inference cost is only slightly higher than that of Reason4Rec.
>
> Given the corresponding improvement in recommendation quality reported in the paper, we consider this modest increase in inference cost acceptable.
>
> Thank you once again for your support and understanding! We will incorporate all the clarifications provided in the rebuttal into the next version. If you have any further concerns, please let us know and we will address them promptly.
>
> Best regards,
> Authors!

---

### Official Review · Reviewer_uhad · 2025-07-09

**Clarity:** 2
**Significance:** 2
**Originality:** 3
**Rating:** 4
**Confidence:** 5

**Summary:**

This paper introduces RecZero, an innovative reinforcement learning (RL)-based approach for recommender systems that leverages large language models (LLMs) to autonomously develop reasoning capabilities for rating prediction tasks. It abandons traditional multi-stage distillation methods, adopting instead a structured, reasoning-guided prompt and rule-based reward modeling via group relative policy optimization (GRPO). A hybrid variant, RecOne, combining supervised fine-tuning (SFT) and RL is also proposed. Extensive experiments validate the performance improvements of both RecZero and RecOne across several datasets.

**Questions:**

See weakness

**Ethical Concerns:**

["NO or VERY MINOR ethics concerns only"]

**Final Justification:**

I raise my score from 3 to 4. The major remaining concern is the real-world application of the proposed RL methods in industrial scenarios given the huge cost and sample inefficiency of RL exploration.

**Limitations:**

See weakness

**Quality:**

2

**Strengths And Weaknesses:**

Strength：
1. Technically sound and well-designed experiments across four diverse and challenging benchmark datasets (Amazon-book, Amazon-music, Yelp, IMDb). Rigorous comparisons with traditional and recent LLM-based recommendation models.
2. Clearly structured and logically coherent. Methods, assumptions, and experimental setups are thoroughly explained.
3. Demonstrates clear improvements over existing approaches, addressing key limitations like static supervision and superficial reasoning capability transfers. The proposed frameworks significantly advance recommendation model capabilities, especially in cold-start scenarios.
4. The use of RL (GRPO) integrated with structured reasoning prompts and hybrid initialization strategies is novel and represents a meaningful departure from standard distillation approaches.

Weeknesses：
1. The motivation is somewhat unclear. In the recommendation search domain, using large models to distill smaller models for recommendation is a reasonable approach, as the smaller model is used for online inference, reducing latency. The motivation raised in the paper are:
    - The teacher model has limited capabilities. However, I think the teacher model is large and powerful, and the student model has not yet fully learned from the teacher model. So "limited capabilities" is not convincing.
    - Calling teacher models are costly. But I think the primary cost is during training, and inference with the smaller model is fast.
    - Superficial Transfer of Reasoning Ability. This point is interesting but lacks experimental evidence to demonstrate that there is a shortcut to transferring reasoning ability.
2. The proposed method does not address the motivation, as it simply shifts the training paradigm from distillation to reinforcement learning (RL), which I believe is not ideal. Firstly, RL does not reduce training time; on the contrary, RL exploration is more complex. Secondly, the paper does not provide experiments proving that RL training is faster than the original distillation approach. Moreover, RL seems like an overcomplicated approach, and in the recommendation domain, there are not many RL-related techniques because they have been less effective over time.
3. The experiments are insufficient. The baseline models are limited, and classic traditional algorithms such as DeepFM, DIN, and SIM are not included, nor are LLM-based recommendation algorithms like TallRec, ReLLa, CoLLM, and RecLoRA. Additionally, important recommendation metrics such as AUC, LogLoss, and NDCG are missing from the evaluation.
4. The writing lacks detail. Algorithms like WDL and RGCL in the baseline are not properly cited, and there is no subsection explaining the baseline models, resulting in a lack of necessary details.

---

> ### Author Rebuttal · Authors · 2025-07-31
>
> We appreciate your comments, which greatly improve our paper. Below we provide the point-topoint responses to address your concerns and clarify the misunderstandings of our proposed
> method.
> > **C1: The motivation is somewhat unclear**
>
> > **C1.1: "The teacher model has limited capabilities" is not convincing.**
>
> Thank you very much for your insightful comments on the motivation of our work. They help us present the paper more clearly.
>
> - **Clarification of Our Presentation:** Although LLMs possess extensive world knowledge and strong cross-domain capabilities, a pronounced domain gap between their pre-training data and the data distribution of recommender systems often causes vanilla LLMs to underperform in real-world recommendation scenarios [1-4]. Accordingly, when we mention that the “teacher model has limited capability,” we are referring to a general-purpose LLM whose abilities are not yet aligned with the recommendation domain. In earlier paradigms, such an unaligned LLM is used as the teacher to analyze user preferences, item features, and matching scores, and these sub-optimal abilities are then distilled into a smaller student model.
>
> [1] TALLRec: An Effective and Efficient Tuning Framework to Align Large Language Model with Recommendation
>
> [2] CoLLM: Integrating Collaborative Embeddings into Large Language Models for Recommendation
>
> [3] LLaRA: Aligning Large Language Models with  Sequential Recommenders
>
> [4] Adapting Large Language Models by Integrating  Collaborative Semantics for Recommendation
> - **Additional Experiment with Structured Reasoning on Rating Prediction:** To further substantiate our claim, we evaluated a vanilla LLM with explicit structured reasoning against SOTA reasoning baselines (see table below). Even with added structure, the LLM shows no clear advantage, confirming that distilling its reasoning traces alone is insufficient for recommendation tasks. This underscores the need to learn task-specific reasoning via RL. We also optimized our code, further boosting RecZero and RecOne.
> || Amazon-Music ||
> |-|-|-|
> ||MAE|RMSE|
> |MF|0.6188|0.8142|
> |WDL|0.6477|0.8597|
> |RGCL|0.6760|0.9103|
> |GPT-4o|0.7438|1.1069|
> |Qwen2.5-7b-Instruct|0.7105|0.9730|
> |Rec-SAVER| 0.6463|0.9262|
> |EXP3RT|0.5608|0.8385|
> |Reason4Rec|0.5442|0.7722|
> |RecZero|**0.4271**|**0.7058**|
> |RecOne|**0.3816**|**0.6776**|
> - **Additional Experiment with a More Refined Method:** We added rating-prediction experiments in which a vanilla LLM with explicit structured reasoning is compared to current reasoning-enhanced baselines. The LLM still shows no clear advantage, confirming that distilling its reasoning traces is inadequate for recommendation tasks. This motivates our RL approach to discover truly useful reasoning. We have also optimized the code, further boosting RecZero and RecOne.
> 1. **RecZero-WithUserTeacher:** replace raw history with the teacher-summarized user preference.
> 2. **RecZero-WithItemTeacher:** replace item meta with the teacher-summarized item feature.
> 3. **RecZero-WithTeachers:** supply both teacher summaries.
> ||Amazon-Music ||
> |-|-|-|
> ||MAE|RMSE|
> |RecZero-WithUserTeacher|0.4663|0.7112|
> |RecZero-WithItemTeacher|0.4519|0.7117|
> |RecZero-WithTeachers|0.4874|0.7094|
> |RecZero| **0.4271**|**0.7058**|
> |RecOne|**0.3816**|**0.6776**|
> > **C1.2: The traditional paradigm that relies on a teacher model is not costly, and inference with the smaller model is fast.**
>
> Your comments prompted us to articulate our motivation more clearly. As discussed in our response to Q1.1, vanilla LLMs are poor teachers for recommendation due to a domain gap. An alternative is to curate a large corpus of human-annotated, recommendation-specific reasoning, yet such manual annotation is prohibitively expensive. RecZero instead learns task-specific reasoning via RL, avoiding this overhead.
>
> We also examined inference efficiency. with the same 7B model size, RecZero attains both the best accuracy and the lowest inference cost versus Qwen-2.5, EXP3RT, and Reason4Rec (see Table C2).
>
> > **C1.3 The claim of a superficial, shortcut-style transfer of reasoning ability is intriguing but presently unsupported by empirical evidence.**
>
> Thank you for your insightful comments on the motivation of our work. Recent studies have investigated how supervised fine-tuning (SFT) influences the generalization of foundation models [1–2].
>
> - SFT Memorizes, RL Generalizes [1] shows that SFT tends to memorize training data and thus generalizes poorly to out-of-distribution scenarios, while RL strengthens a model’s core abilities and enhances domain generalization.
>
> - SFT or RL? [2] systematically compares SFT and RL and reveals that SFT often leads to the imitation of “pseudo reasoning paths’’ generated by an expert model. These paths may look similar to the native reasoning produced by RL-trained models, yet they usually contain redundant, hesitant, or low-information steps and sometimes even incorrect reasoning. Moreover, our earlier response (C1.1) has demonstrated that the expert model used in recommendation tasks is itself sub-optimal.
>
> [1] SFT Memorizes, RL Generalizes: A Comparative Study of Foundation Model Post-training
>
> [2] SFT or RL? An Early Investigation into Training R1-Like Reasoning Large Vision-Language Models
>
> Owing to space limitations, we discuss the case study only in our response to Reviewer dSSq’s Comment C3. We kindly refer you to that section.
>
> > **C2: Simply replacing distillation with a more complex RL paradigm fails to address the motivation, offers no evidence of faster training, and is questionable given RL’s historically limited success in recommendation tasks.**
>
> Based on your suggestions, we have made clarifications and additional experiments:
> - **Clarification of Our Training Cost:** Concerning your comment that we simply replace distillation with RL and that RL exploration is more complex and not shown to shorten training time, we note that, although RL may consume more computation concurrently, it can achieve a noticeable performance gain after only a small amount of data compared with SFT. This means that, given the same computational budget, RL needs far fewer training iterations to reach performance on par with SFT.
> - **Additional Experiment on Training and Inference Cost:** we report the training and inference costs of several LLM-based reasoning-enhanced rating-prediction methods in Table below, showing that RecZero (early-stop) matches the performance of previous paradigms using only 0.48 K training samples and 0.4 hours of computation, while requiring significantly fewer inference tokens than multi-stage reasoning baselines.
>
> |Method|Samples to Converge|Trained Models|Total Training Time|Avg. Inference Tokens|Inference Stages|Total Inference Tokens|
> |---|---|---|---|---|---|---|
> |qwen|\-|\-|\-|399.92|1|399.92|
> |EXP3RT|20K|3|1.2h|187.63|3|562.89|
> |Reason4Rec|20K|3|1h|175.49|2|350.98|
> |RecZero(early-stop)|**0.48k**|**1**|**0.4h**|**331.64**|**1**|**331.64**|
> |RecZero(full)|**1.2K**|**1**|**1.1h**|**310.52**|**1**|**310.52**|
>
> - **Additional Analysis on Method Complexity and Effectiveness:** We appreciate your expert insight into recommender systems. As you noted, RL delivered only modest gains to recommendation tasks prior to the LLM era. However, the recent emergence of LLMs with rich world knowledge, cross-domain skills, and intrinsic reasoning ability has made RL-based exploration substantially more effective, as evidenced by recent studies[1–2]. To address your concern about potential complexity, we juxtapose earlier reasoning-enhanced rating-prediction frameworks with our RecZero in Table below, which shows that RecZero already simplifies the pipeline considerably.
>
> |Method|Trained Models|Training Stages|Teacher-Free|End-to-End|
> |-|-|-|-|-|
> |Rec-SAVER|1|1|❌|✅|
> |EXP3RT |3|3|❌|❌|
> |Reason4Rec|3|3|❌|❌|
> |RecZero|1|1|✅|✅|
>
> [1] Rec-R1: Bridging Generative Large Language Models and User-Centric Recommendation Systems via Reinforcement Learning
>
> [2] Reinforced Latent Reasoning for LLM-based Recommendation
>
>
> > **C3: The experimental evaluation is inadequate.**
>
> You have raised a highly general question. At present, our RecZero and RecOne models are designed for the rating-prediction task in recommender systems, where the standard evaluation metrics are MAE and RMSE. Following your suggestion, we have adapted our method to other tasks.
> - **Additional Experiment on CTR Prediction:** For the CTR-prediction task, we adopted the same Amazon-Book dataset configuration as CoLLM, treating ratings ≥ 3 as positive (clicked) and the rest as negative. The results are reported below.
> ||AUC|UAUC |
> |-|-|-|
> | MF | 0.7134 | 0.5565 |
> | LightGCN | 0.7103 | 0.5639 |
> | SASRec | 0.6887 | 0.5714 |
> | DIN | 0.8163 | 0.6145 |
> | CTRL(DIN) | 0.8202 | 0.5996 |
> | ICL | 0.4820 | 0.4856 |
> | Soft-Prompt | 0.7224 | 0.5881 |
> | TALLRec | 0.7375 | 0.5983 |
> | CoLLM-DIN | 0.8109 | 0.6225 |
> | RecZero | **0.8330** | **0.6301** |
>
> - **Additional Experiment on Sequential Recommendation:** Following your suggestion, we have extended our framework to a sequential-recommendation setting and report both HR and NDCG on the Amazon-Music dataset, as shown below.
>
> | |HR@5|HR@10|NDCG@5|NDCG@10|
> |-|---|---|---|---|
> |Caser|0.0103|0.0110|0.0137|0.0144|
> |GRU4Rec|0.0165 |0.0189|0.0133|0.0140|
> |SASRec|0.0174|0.0200|0.0144|0.0163|
> |BigRec|0.0050| 0.0107 |0.0031|0.0049|
> |RecZero|**0.0235**|**0.0390**|**0.0148**|**0.0198**|
> > **C4: The manuscript lacks sufficient detail.**
>
> Thank you for highlighting the missing details regarding our baseline settings. We will (i) add the appropriate citations for Wide & Deep-Learning and Relational Graph Contrastive Learning, and (ii) introduce a new subsection entitled “Baseline Models,” in which we summarize the architecture, hyper-parameters, and training procedure of every baseline used in our study.

---

> > ### Comment · Reviewer_uhad · 2025-08-04
> > **Response From Reviewer**
> >
> > Regarding my concerns about the training complexity, the authors claim that the proposed RecZero is more efficient than existing RL methods. But my major concern is that the performance gain from RL exploration is not worthy. In RS, it is more important to incorporate more training data with sufficient features and real-time feedback with fast SFT. Iteratively exploring the same data corpus can lead to better performance, but not in an industry-friendly way.

---

> ### Author Response · Authors · 2025-08-05
>
> We sincerely thank for the follow-up comments. Your major concern is that “the performance gain from RL exploration is not worthy”. Below we present new experiments and a clearer analysis showing why RecZero and RecOne remain attractive.
>
> We follow the same experimental settings as in our previous experiments and add tow extra variants: (1) RecOne (SFT) – the RL phase of RecOne is removed and the model is trained to full convergence; (2) RecOne(full) – original SFT → RL pipelin.
>
> |Method|Paradigm|MAE⬇️|RMSE⬇️|Samples×Epochs to Converge|Trained Models|Total Training Time|Avg.InferenceTokens|InferenceStages|Total Inference Token|
> |---|---|---|---|---|---|---|---|---|---|
> |Qwen|\-|0.6821|0.9796|\-|\-|\-|399.92|1|399.92|
> |EXP3RT|SFT|0.5608|0.8385|20K|3|1.2h|187.63|3|562.89|
> |Reason4Rec|SFT|0.5442|0.7722|20K|3|1h|175.49|2|350.98|
> |RecZero(early-stop)|RL|**0.5419**|**0.7736**|**0.48k**|**1**|**0.4h**|**331.64**|**1**|**331.64**|
> |RecOne(SFT)|SFT|0.6472|0.9284|**20K**|**1**|**0.6h**|**597.32**|**1**|**597.32**|
> |RecZero(full)|RL|**0.4271**|**0.7058**|**2.4K**|**1**|**1.1h**|**310.52**|**1**|**310.52**|
> |RecOne(full)|SFT+RL|**0.3816**|**0.6776**|**2.6K**|**2**|**1.4h**|**412.79**|**1**|**412.79**|
>
> **1. Larger performance gain per unit cost**
> - Pure SFT versus pure RL. RecOne (SFT) needs 20 K labels, 0.6 h GPU time and 597 inference tokens to reach MAE = 0.6472. RecZero (early-stop) beats it (MAE = 0.5419) with 97.6 % fewer labels (0.48 K), 33 % less training time (0.4 h) and almost half the serving cost (331 tokens).
> - Deeper RL training. With exactly the comparable inference length and GPU time, RecZero (full) further lowers MAE/RMSE to 0.4271/0.7058, out-performing the strongest pure-SFT baseline Reason4Rec by 21.5 % MAE reduction while consuming only 12 % of its labels (2.4 K vs 20 K).
> - In terms of "error reduction per 1 K labels", RecZero is over 5 × more label-efficient than SFT.
>
> **2. Fast adaptation to cold-start or drifting slices**
> - Modern RS platforms face thousands of new items and contexts each day. RecZero can be re-optimised with a few-hundred live interactions collected in minutes, whereas SFT must wait until a large, curated batch is available. This few-shot adaptation is difficult for data-hungry SFT alone.
>
> **3. Simplest training pipeline among reasoning-enhanced methods**
> As Table in the rebuttal shows, RecZero keeps
> - one model,
> - one training stage,
> - no distillation teacher,
> - end-to-end optimisation.
> So the engineering overhead is strictly smaller than prior multi-stage SFT pipelines such as EXP3RT/Reason4Rec (3 stages & models).
>
> **4. Complementary, not competing, to SFT**
>
> We do not claim RL should replace SFT in all cases. RecOne (full) shows that an SFT warm-start followed by a light RL pass pushes MAE/RMSE even further to 0.3816/0.6776. Therefore an operator can first run a fast SFT pass on abundant public logs and then invoke RecOne-style RL on the latest or strategically important traffic.
>
> To summarise:
> - Pure SFT loses to pure RL. With the same backbone, RecOne (SFT) needs 20K labels, longer training, and almost twice the serving tokens, yet still under-performs RecZero (early-stop), which uses only 0.48K labels and 0.4h GPU time.
> - RecZero(full) achieves a further 21.5% MAE reduction over the strongest SFT baseline at a comparable computational cost.
> - RecZero and RecOne adapt to fresh or sparse data within minutes.
> - The pipeline remains single-model, single-stage, and teacher-free—simpler than previous reasoning-enhanced approaches.
>
> We hope these clarifications address your concern.

---

> > ### Comment · Reviewer_uhad · 2025-08-07
> > **Response From Reviewer**
> >
> > I appreciate the authors' response. Although I am still concerned about the real-world application of RL with LLMRec, I will raise my score to 4, since my major concerns are already addressed.

---

> > > ### Author Response · Authors · 2025-08-07
> > >
> > > Thank you very much for your positive feedback. We sincerely appreciate your constructive suggestion and will definitely add this in the revised version. Your valuable comments have helped us improve our work significantly. Thank you again for your support.

---

### Decision · Program_Chairs · 2025-09-17

**Decision:**

Accept (poster)

**Comment:**

Existing distillation-based methods for using LLMs in recommendation suffer from the teacher model's limited capability, costly supervision, and superficial transfer of reasoning skills. This paper proposes a novel reinforcement learning (RL) paradigm that trains a single LLM autonomously for rating prediction via structured "Think-before-Recommendation" prompts and a rule-based reward model (GRPO) to optimize reasoning trajectories. Experimental results show the effectiveness of the proposed model. The rebuttal has addressed the concerns of the reviewers, and all the reviewers agree to accept this work.